# Grazing Land Productivity, Floral Diversity, and Management in a Semi-Arid Mediterranean Landscape

Georgios Psyllos [1,*], Ioannis Hadjigeorgiou [2], Panayiotis G. Dimitrakopoulos [3] and Thanasis Kizos [1]

1 Department of Geography, University of the Aegean, University Hill, 81100 Mytilene, Greece; akizos@aegean.gr
2 Department of Animal Science, Agricultural University of Athens, Iera Odos 75, 11855 Athens, Greece; ihadjig@aua.gr
3 Department of Environment, University of the Aegean, University Hill, 81100 Mytilene, Greece; pdimi@aegean.gr
* Correspondence: g.psyllos@geo.aegean.gr

**Abstract:** Most grazing lands in Mediterranean ecosystems that support extensive sheep farming systems are characterized by unfavorable edapho-climatic conditions, especially in semi-arid areas. Often, though, their use is far from sustainable, causing erosion and ecosystem degradation impacts. In this paper, we explore the use, productivity, and flora diversity of typical Mediterranean grazing lands in four farms at the Agra locality in the western part of Lesvos Island, Greece. For a period of two consecutive growing seasons (September to June), we recorded herbage biomass on 16 plots of grazing lands with three measurements per season of land cover and plant productivity (biomass) inside small exclosures (cages) protected from grazing. We recorded the species richness of herbaceous plant communities within and outside the cages at the end of every growing season, the period of maximum growth of herbaceous species. We also chemically analyzed the biomass for crude protein at the end of each season. Results show sizable productivity differences among pasture plots as well as seasons and an overall medium to high degree of productivity and species richness considering the relatively intensive grazing, with little differences over the different cages and the degree of grazing intensity. These results suggest that the "history" of the fields is important, as grazing lands that had been used for arable crops in the past, as well as those leveled and in favorable locations, were the most productive and diverse ones, while shallower soils and inclined grazing lands showed signs of overuse and degradation. Overall, though, these ecosystems showed a high degree of resilience despite their intensive use.

**Keywords:** grazing lands; semi-arid Mediterranean; floral diversity; biomass measurements; Lesvos Island

## 1. Introduction

Humans, pastures, and grazing animals are closely related factors in Mediterranean ecosystems. The dynamic coexistence of these ecosystems and human societies has determined and shaped the evolution and stability of both over several millennia [1].

Pastoral land in the Mediterranean basin covers an estimated area of 850,000 square kilometers, mostly occupying areas characterized by unfavorable pedo-climatic and soil conditions. Their use is limited mainly to extensive systems or is bounded to specific periods of the year, when animals have low nutrient requirements [2–4]. However, such ecosystems play a crucial role in sustaining local societies and their economies in marginal lands of the Mediterranean through livestock farming and in particular sheep husbandry [5–7]. In these areas, low-intensity and site-specific agricultural practices, mainly based on grassland resources, have evolved, over centuries, to limit risks associated with the inter- and intra-annual climatic fluctuations and to ensure regular production [4,8,9].

Traditional practices carried out in these areas are often regarded as environmentally friendly and landscape-preserving, and the lands are also considered as being of high nature value [10,11]. Mediterranean grassland-based systems are usually extensive, with low use of agrochemicals and irrigation, while they are utilized predominantly by small ruminants due to their high efficiency in the use of locally available feeding resources [12,13] as well as their adaptation to the specific environments of these areas [14,15]. These farming systems have proved to be resilient to frequent, but moderate, disturbances such as deforestation, periodic fires, and overgrazing, by developing strategies to optimize the production of multiple goods as well as ecosystem services [16,17].

The productivity of Mediterranean grazing lands is limited by physical constraints: such as climatic and soil characteristics [9]. Their growing season ranges from 4 to 8 months, depending on precipitation (300–1000 mm/year) and timing as well as the tolerance of flora to water scarcity. Annual and interannual forage production under rainfed conditions is often highly variable, although generally low, depending on land management and soil fertility. Typically, average dry matter yields range from 0.5–1.0 t ha$^{-1}$ year$^{-1}$ in semi-natural lands found on marginal soils, to 6.0–7.0 t ha$^{-1}$ year$^{-1}$, in agriculturally improved grasslands [18]. In semi-natural grasslands, forage is usually of low nutritional quality, often worsened by the presence of plant species with anti-nutritional properties or other traits that limit their acceptability for ruminants. Dry matter accumulation ranges between 110 kg ha$^{-1}$ day$^{-1}$, in the most favorable season (spring), down to 20 kg ha$^{-1}$ day$^{-1}$ in autumn [19,20]. Annual species dominate in the herbage, but most of them are also encroached by perennial species such as *Cistus ladanifer* [21], *Genista acanthoclada*, *Sarcopoterium spinosum* [22], *Phlomis fruticosa* [23], etc., that can contribute to some extent to feed resources, but are of limited and unbalanced nutritional value. An increase in shrub cover leads to a decline in herbage production, as well as an overall reduction in the nutritional value of the land's forage production [24].

Livestock grazing impacts grazing lands' biodiversity and in particular plant community composition, as well as quantity and quality of the herbage produced [25], vegetation dynamics [26], species and bio-societies variability [27], and the landscape overall [28]. In total, grazing activity contributes to a rich mosaic of vegetation [29] and results in the creation and preservation of all biodiversity forms [30,31]. However, changes in plant community diversity, created by grazing animals, can vary with environmental conditions, including regional variation in climate [32], the evolutionary history of grazing [33], as well as the supply of nutrients [34]. Although Mediterranean semi-arid pasturelands have proven to be very resilient over the ages and have managed to recover from frequent and intense disturbance events such as fires, droughts, and the constant exploitation of humans, nowadays they appear to have reached a critical point [35].

Lesvos Island, in the eastern Mediterranean, is the third-biggest island in Greece and the seventh in the Mediterranean, occupying an area of about 163,280 hectares [36]. Land cover varies significantly from west to east, the eastern part is covered by olive groves, while the western is characterized by phryganic vegetation and used as pastoral land. Western Lesvos is a typical semi-arid Mediterranean area with a sparse population and low development, combined with environmental sensitivity and serious problems of local degradation and desertification [36]. Agra, a settlement in western Lesvos has a long tradition of sheep farming and during recent decades sheep farmers have shifted from traditional farming practices towards modernization, leading to intensification of production [37]. In fact, the number of farmed sheep there increased by 297.5%, between 1961 and 2010, while the number of holdings decreased slightly by 4.9% [38]. These changes have led to the average size of a holding increasing by 318.1% from 34.5 sheep heads up to 144.3 during the same period. These developments coincided with a shift of farming systems in the area towards importing considerable quantities of supplementary feeds, in an attempt to maintain the balance between the animals' nutritional requirements and the available local feed resources. It is questionable though as to whether management practices, in the use of this resource, are sustainable in terms of herbage productivity, forage

quality, and land maintenance. In this study, we aimed to assess the effects of sheep farming practices on the sustainability of a semi-arid Mediterranean pastoral landscape; thus we recorded several characteristics of the land, including land cover, herbaceous vegetation productivity, plant diversity, and herbaceous feed quality over two seasons. Grazing practices, pasture improvement practices, and climatic conditions were also considered. The overall outcome will serve as a guide for management practices and policies for these pastoral lands.

## 2. Materials and Methods

### 2.1. Study Site

The study site of Agra is located on the western part of Lesvos Island (Figure 1) with approximately 1000 people and 200 sheep farms, where the terrain is hilly with steep to moderate slopes of acid volcanic rocks, and the soils are stony and shallow. Land cover is dominated by shrublands and phrygana vegetation [39] amounting to 76% of the area, while the rest comprises 17% olives and oak trees plots and 6.7% alluvial plains cultivated with annual crops [40]. The climate is characterized by an annual rainfall of 415 mm and a mean annual temperature of approximately 17 °C, with an average range between 12.2 °C and 21.4 °C [41]. Soils in the area are classified as shallow Typic Xerochrepts or Lithic Xerochrepts or Typic Xerofluvents (recent alluvial soils) and Typic Haploxeralfs. The soils of the pastures used here are described as shallow (15–30 cm) or medium-deep (30–60 cm) and are composed of lava and tuff material in medium to heavy incline (6–12% and more than 12%). These pastures present rocky parent material at depths from 20 to 100 cm. Only three former arable fields are deeper than 60 cm, almost flat (0–2%), and made from alluvial material.

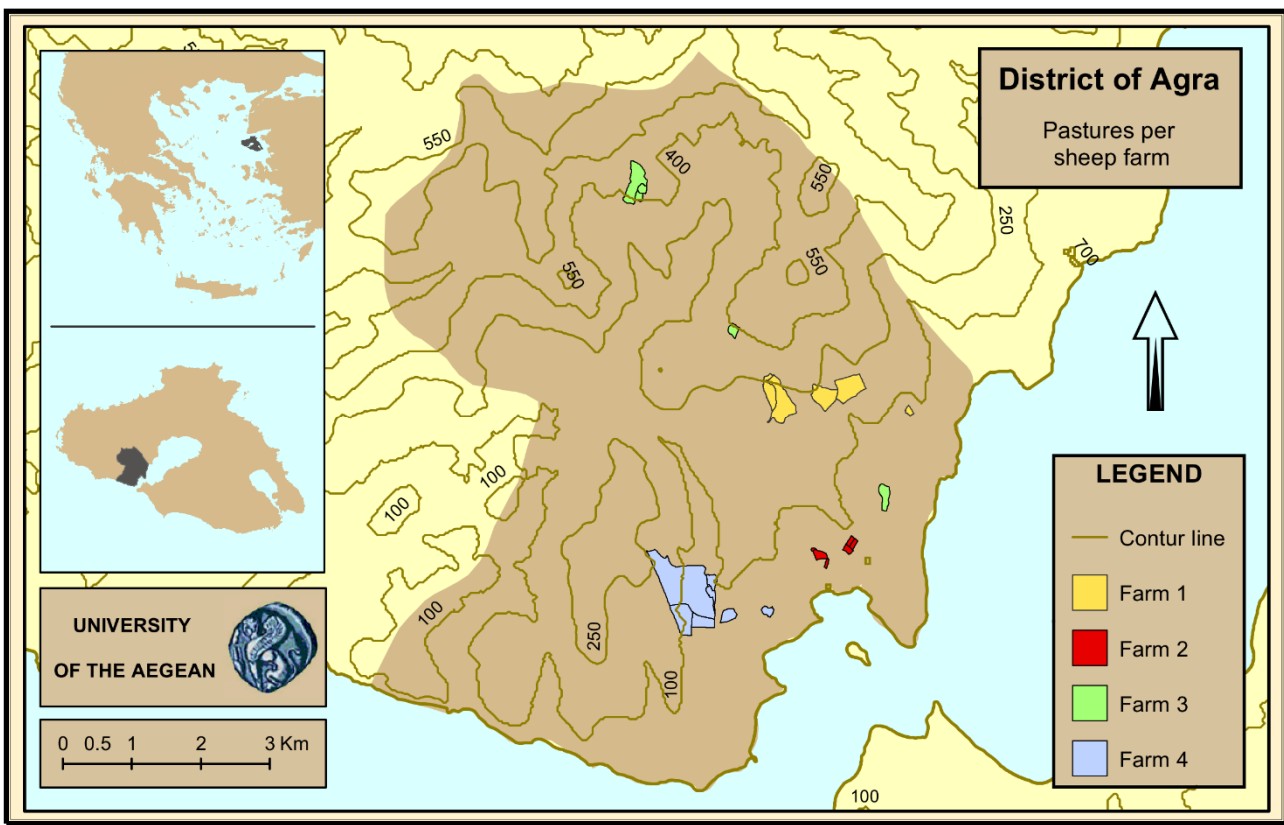

**Figure 1.** Case study area: Western Lesvos (Agra) (Prepared by authors).

*2.2. Research Methods*

For a period of two consecutive growing seasons (September to June), we recorded

1. herbage biomass;
2. land cover;
3. species richness of herbaceous plant communities;
4. chemically analyses of the biomass for ash, crude protein, and crude fiber.

Field measurements were performed on land utilized by four sheep farms (Table 1), which were selected to represent typical cases in size and practices in the area, thus excluding very small as well as very intensive ones (animals kept only indoors) and choosing farms exclusively raising sheep of the local breed of Lesvos [42]. These farms utilized a total of 14 individual fenced pasture plots, managed autonomously, which represented different types of terrain, from steep hilly land to former arable plots in the lowlands. Nine of these pastures were categorized as "improved" (pastures 1.1, 1.2, 2.1, 2.2, 2.3, 3.1, 3.2, 3.3, and 4.1 representing pastures in which farmers have performed some kind of improvement actions, such as maintaining terraces, removing rocks and undesirable plants, including cultivation of arable plants in the past) and the rest of the pastures were categorized as "undisturbed". Meteorological data were collected from Sigri Station covering a decade from 2006 to 2016, including precipitation and temperature.

**Table 1.** The intensity of production for the selected farms (2013) and the corresponding median for the farms of the study area (data from the Lesvos branch of the National Milk Organization).

|  | **Sheep** | **Milk (kg)** | **Intensity of Production (Milk/Sheep/Year)** |
|---|---|---|---|
| Farm 1 | 337 | 56,270 | 167.0 |
| Farm 2 | 248 | 27,832 | 112.2 |
| Farm 3 | 179 | 28,892 | 161.4 |
| Farm 4 | 398 | 38,940 | 97.8 |
| Study area (median) | 184.5 | 15,771 | 85.2 |

Biomass production was measured at least on one sampling station for each of the pasture plots, depending on its size, the diversity of the terrain, and the land cover. The location of each sampling station was selected on the basis that no bare soil was present, the surface was free of stones and perennial vegetation, and there was enough space to accommodate the required number of herbage protection cages. The cages were made with wire mesh (mesh size $1 \times 1.2$ cm), covering a circular area of 0.25 m$^2$ and with a height of 0.7 m covered with a wire mesh lid. To anchor the cages, three iron rods were driven into the ground. The first two cages were placed at the start of the vegetative period, in October (Figures 2 and 3). One of them ("control cage") was sampled only at the end of the growing period (control cage). The second ("1st cage") was harvested at ground level three months later (December) and a second time at the end of the season along with all cages, while a new cage ("2nd cage") was placed nearby. During the second sampling, in early March, the 2nd cage, placed in December, was harvested and a new cage ("3rd cage") was also placed. During the third sampling, from late May to early June, when herbage was almost dry, all four cages were harvested (the "1st cage" and the "2nd cage" for the second time). To record the standing herbaceous biomass in the plot during each sampling session, a sample was harvested from an area of 0.25 m$^2$ representative of the overall field vegetation ("grazed area sample") [43]. The final sampling was made at different periods each year due to the weather conditions that matured plants earlier in the second year: in the first year, this was in early June, while in the second in mid-May. After the completion of every sampling, the samples were dried in paper bags at 60 °C in an oven for at least twenty-four hours.

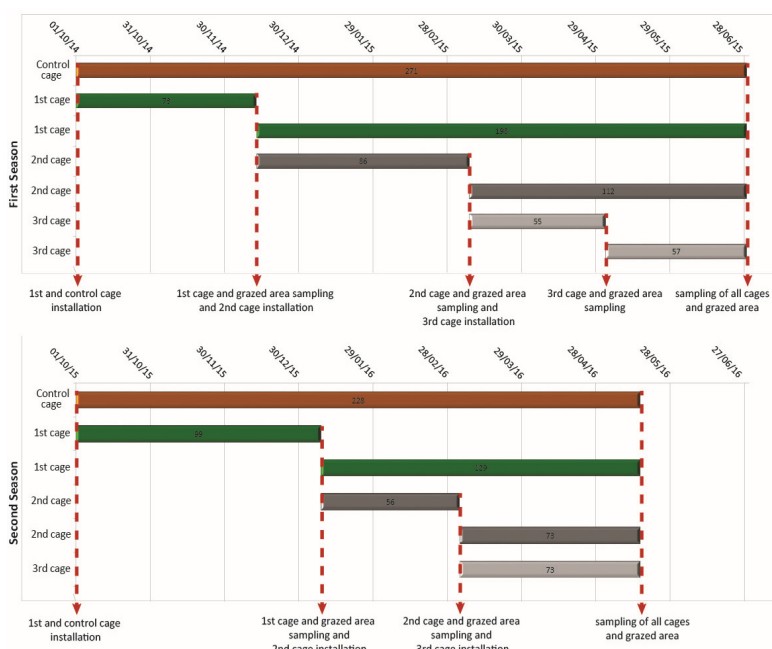

**Figure 2.** Timeline of field measurements.

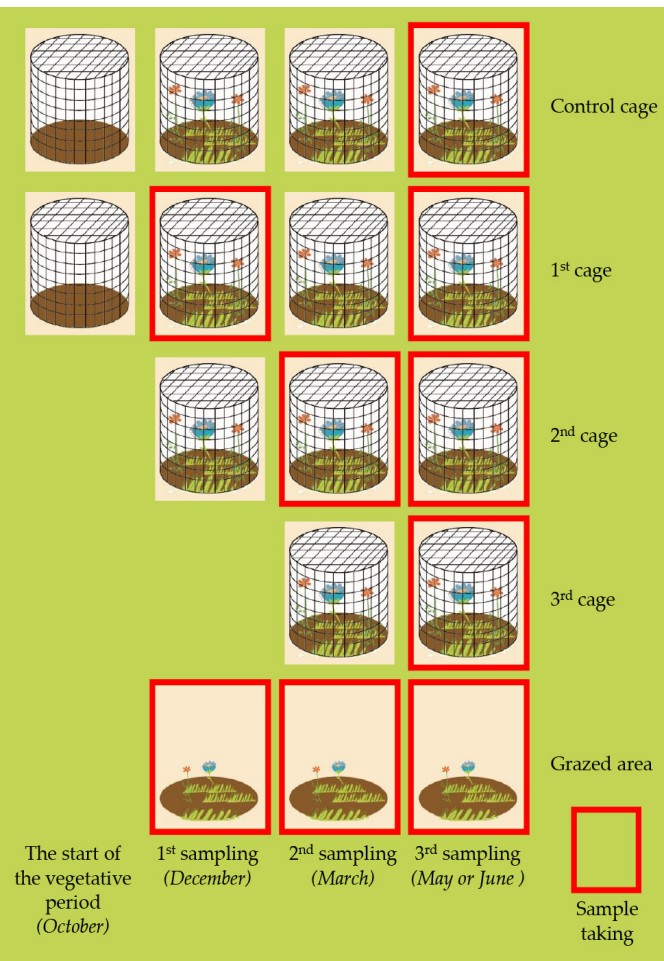

**Figure 3.** The sampling method: number of samples taken per year: control cage 1, 1st and 2nd cage × 2 = 4, 3rd cage 1, grazed area 3, in total 9.

Land cover was recorded in each study year as perennials (separately for some species, i.e., thorny burnet (*Sarcopoterium spinosum*) and branched asphodel (*Asphodelus ramosus*)), annuals, rocks, litter, and bare soil by the step-point method [44]. Ground cover was noted down for 364 points in nine different periods for both years, situated at 1-step intervals in a 25-step transect perpendicular to the slope and another 25-step transect parallel to contours. The two transects cross at the center of the plot. Cover data are expressed in percentages.

Plant diversity was assessed at the end of each sampling season using the samples of the last harvest for all cages. Examined specimens were classified at family level and species, when this was feasible, as some were eaten or had partially withered. We used [45,46] for the nomenclature of taxa.

Herbage nutritional quality was assessed on dried material of all cages from the final samplings at the end of the two years. Samples were pre-ground on a hammer mill to pass a 6 mm sieve and then subsampled and finely ground on a laboratory mill (CT 193 CyclotecTM FOSS, Hilleroed, Denmark) using a 1 mm screen. Analyses were conducted for moisture (AOAC method 930.15), crude protein (CP) by the Kjeldahl method (AOAC method 984.13) on a 2300 Kjeltec Analyser Unit Foss Tecator, ash (Ash) by ashing overnight at 550 °C (AOAC method 942.05), and crude fiber (CF) (AOAC 978.10) on an ANKOM 220.

### 2.3. Analysis

Descriptive statistics were used to explore the variables used. For all comparisons, we used analysis of variance (ANOVA). First, we tested for differences in plant productivity per regrowth day between different types of cages for each pasture for both years. Additionally, we examined the effect of different types of cages and years on the number of recognized species, the number of different plant families (grass, legume, non-leguminous herb), and the crude protein in plant biomass. The degree of grazing intensity was calculated as the number of sheep per hectare per pasture.

The variables used are:

1. Type of pasture: shows if the pastures are undisturbed or if there have been actions by farmers to improve them. The improvement of grazing land is by making terraces, removing rocks and undesirable plants, this usually stands for the land which has changed use, most of the time from arable land to pastures.
2. Land cover classes: the land cover was recorded in five classes as % of total cover: bare soil, annuals, rocks, *S. spinosum*, *Asphodelous ramosus*.
3. Productivity (kg dm/ha): expresses the productivity of herbaceous dry mass per hectare as calculated by multiplying the biomass produced by each cage by the percentage of herbaceous biomass coverage of each pasture. This productivity was assumed to be homogenous for the pastures in the cases of a single sampling cage, while in pastures with more than one sampling point, the average productivity of all sampling points was used.
4. Productivity per day (regrowth days) ((kg dm/ha)/day): Productivity/Regrowth days: expresses the productivity of herbaceous dry mass per hectare and per day for the period that each sampling represents (all days appear in Figure 2).
5. Grazing pressure ((animals/ha)/day): grazing practices have been provided by the farmers for the whole grazing period as the adult animals that grazed a pasture for 15-day intervals. This implied that the shortest grazing period for each pasture is 15 days and up to the whole period.
6. Dry matter/sheep/day ((kg dm)/day): expresses the dry matter that was available to the animals that were on the pasture per day for the days that the pasture was grazed.
7. Number of different species (N): expresses the total number of the different species recognized, classified in families, and more general to grasses, legumes, and other families.
8. Herbage nutritional quality is measured as % of total dry matter content in proteins, ash, and crude fiber.

## 3. Results

### 3.1. Precipitation and Land Cover

Precipitation in the two growing seasons differed since, in the first hydrological year of the measurements, which coincides with the biomass growing season, total precipitation was 797.3 mm, substantially higher than the historical average annual rainfall of 464.6 mm in the area and that of the total maximum value of the nine years of measurements available, while in the 2nd year total precipitation was 353.6 mm, close to the minimum of the nine recent years (322.1 mm). Out of the maximum rainfall of the first year, though, 45% amounted to a period of two months, from late December to late February (Figure A1). This rainfall was higher than that of the whole second hydrological year. The timing in relation to the biomass growing season and the intensity of the rainfall suggest that the effect on biomass is not as important as absolute values imply. Rainfall in the second hydrological year was evenly distributed within the season and the amount that fell in April and May was equal to that of the first year. Moreover, the average annual temperature for the last 9 years was 17.4 °C, very similar to the average of the two years of the experiment.

Land cover varies within the season. Bare soil decreases from one quarter to one-tenth of the total area as annuals grow after the end of winter, covering 57% of the total area at the end of the season. Rocks and *S. spinosum* cover did not differ significantly within the season (Figure 4). High levels of variability were recorded among different pasture plots, with pastures completely dominated by annuals (typically flat and "improved" plots) on the one hand, while others were dominated by *S. spinosum* and/or rocks (Figure 4). These differences are reflected in the grazing practices and especially in the pressure applied by the farmers, which appear to broadly regulate the grazing pressure according to the characteristics of the pasture.

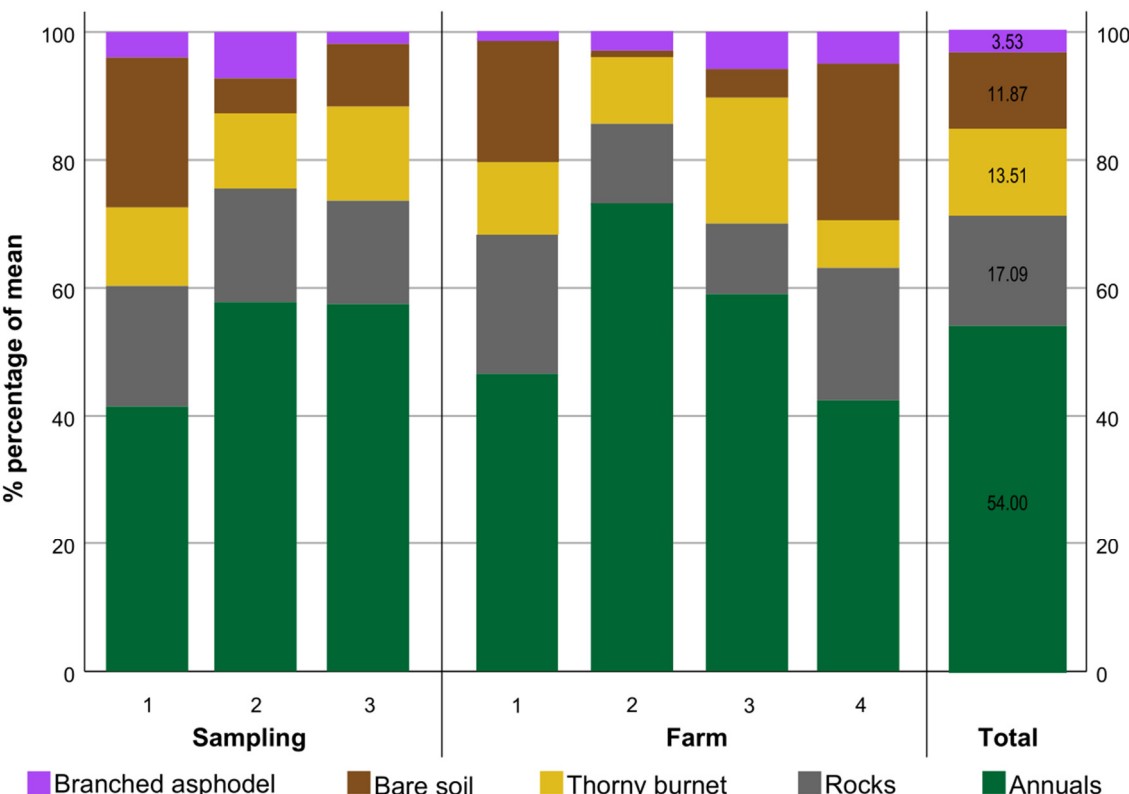

**Figure 4.** Land cover in five groups (annuals, bare soil, thorny burnet, rocks, and asphodel) per sampling period and farm.

### 3.2. Biomass Production, Productivity, and Grazing Practices

Herbage productivity (kg DM/ha) varies significantly between pastures, ranging from very small values close to 300 kg/ha to those of about 5800 kg/ha at a flat pasture plot, which in the past was used for annual crops (Table 2). Similarly, DM production per hectare and day differed significantly between plots, although these differences were slightly lower than total productivity. Moreover, there were statistically significant differences between the cages established at different dates for both years (Table 3): first season one-way ANOVA $F_{(4,121)} = 2.74$, $p < 0.05$; second season one-way ANOVA $F_{(4,121)} = 3.28$, $p < 0.01$; for both seasons one-way ANOVA $F_{(4,247)} = 5.59$, $p < 0.05$.

**Table 2.** Productivity and management indicators of pastures, the counts represent the number of repetitions (samplings), 9 for each year (control cage 1, 1st and 2nd cage × 2 = 4, 3rd cage 1, grazed area 3), 18 in total.

| Pasture Indicators | | | Productivity (kg dm/ha) | | Productivity/Regrowth Days (kg dm/ha)/Day | | Grazing Pressure (Animals/ha)/Day | | Dry Matter/Animal/Day (kg dm)/Day | |
|---|---|---|---|---|---|---|---|---|---|---|
| Pasture | Count | Mean | Standard Error of Mean | Mean | Standard Error of Mean | Mean | Standard Error of Mean | Mean | Standard Error of Mean | |
| 1.1 | 18 | 1249.27 | 276.62 | 10.90 | 1.93 | 5.25 | 0.63 | 3.35 | 0.84 |
| 1.2 | 18 | 416.10 | 60.32 | 4.24 | 0.71 | 11.05 | 02.06 | 0.63 | 0.17 |
| 1.3 | 18 | 570.28 | 142.38 | 4.65 | 0.83 | 1.76 | 0.25 | 6.65 | 2.22 |
| 1.4 | 18 | 274.06 | 79.45 | 2.32 | 0.70 | 13.16 | 0.57 | 0.17 | 0.05 |
| 2.1 | 18 | 2161.01 | 485.67 | 18.80 | 3.53 | 5.30 | 0.48 | 4.80 | 1.21 |
| 2.2 | 18 | 5824.86 | 1273.61 | 49.69 | 9.30 | 43.33 | 2.12 | 1.13 | 0.20 |
| 2.3 | 18 | 1019.10 | 230.22 | 08.07 | 1.11 | 36.20 | 3.79 | 0.34 | 0.08 |
| 3.1 | 18 | 1191.02 | 226.69 | 12.51 | 2.71 | 4.37 | 1.51 | 1.13 | 0.34 |
| 3.2 | 18 | 1360.27 | 306.53 | 11.65 | 2.14 | 9.70 | 02.04 | 1.68 | 0.64 |
| 3.3 | 18 | 2124.49 | 468.53 | 17.45 | 2.93 | 10.06 | 1.23 | 2.71 | 0.80 |
| 4.1 | 18 | 1211.06 | 283.37 | 9.75 | 1.50 | 62.82 | 3.66 | 0.16 | 0.02 |
| 4.2 | 18 | 1594.38 | 301.02 | 14.78 | 2.83 | 11.09 | 2.28 | 1.70 | 0.33 |
| 4.3 | 18 | 351.26 | 87.35 | 3.14 | 0.54 | 1.39 | 0.23 | 2.50 | 0.68 |
| 4.4 | 18 | 320.72 | 93.89 | 03.06 | 0.81 | 01.04 | 0.24 | 2.40 | 0.67 |
| 1st season | 126 | 1250.23 | 199.45 | 9.18 | 1.35 | 16.07 | 1.75 | 1.63 | 0.42 |
| 2nd season | 126 | 1559.46 | 199.16 | 15.25 | 1.69 | 14.86 | 1.74 | 2.66 | 0.32 |
| Total | 252 | 1404.85 | 140.99 | 12.22 | 1.10 | 15.47 | 1.23 | 2.15 | 0.27 |

Grazing pressure follows productivity patterns, as clearly farmers recognize the pastures that provide more biomass and graze them more heavily and/or for a longer time, resulting in a much more balanced distribution of values (Table 2). Nevertheless, differences remain and suggest different grazing practices and different dependence on supplementary feed to cover the animals' nutritional requirements. These differences are evident in the comparison between improved and undisturbed pastures: the daily productivity is three times greater (15.9 kg DM/ha in the improved on average, compared to 5.5 kg DM/ha for undisturbed ones, statistically significant differences, *t*-test = −4.687, $p < 0.01$), and although slightly fewer plants were recognized on average in improved pastures (10.8 compared to 11.7 species in undisturbed ones), many of these species were legumes (1.2 on average in improved compared to 0.8 in undisturbed pastures, the differences are not statistically significant).

**Table 3.** Dry mass productivity per ha per days of regrowth (dm kg/(ha*reg. days)) The differences are statistically significant: first season one-way ANOVA $F_{(4,121)} = 2.74$, $p < 0.05$; second season one-way ANOVA $F_{(4,121)} = 3.28$, $p < 0.01$; for both seasons one-way ANOVA $F_{(4,247)} = 5.59$, $p < 0.05$.

|  | Type of Cage | Count | Mean | Std. Deviation |
|---|---|---|---|---|
| | Control cage | 14 | 10.83 | 11.36 |
| | 1st cage | 28 | 10.22 | 13.63 |
| 1st season | 2nd cage | 28 | 13.07 | 21.78 |
| | 3rd cage | 14 | 15.11 | 22.18 |
| | Grazed area | 42 | 3.38 | 3.62 |
| | Control cage | 14 | 17.17 | 11.53 |
| | 1st cage | 28 | 16.45 | 25.24 |
| 2nd season | 2nd cage | 28 | 16.51 | 16.44 |
| | 3rd cage | 14 | 28.34 | 24.02 |
| | Grazed area | 42 | 8.60 | 12.89 |
| | Control cage | 28 | 14.00 | 11.68 |
| | 1st cage | 56 | 13.33 | 20.34 |
| 1st and 2nd season | 2nd cage | 56 | 14.79 | 19.20 |
| | 3rd cage | 28 | 21.72 | 23.67 |
| | Grazed area | 84 | 5.99 | 9.77 |

The differences in the average values of productivity per ha per day are statistically significant for the different cage types (Table 3). Average daily values in late spring (3rd cage) are much higher than all other periods and from the overall averages and this pattern stands for all individual pastures. The lowest values are recorded in the grazed area. The average values of the rest of the cages are relatively close. Significant differences are evident between the two seasons in all cages and grazed areas. The average productivity per day for the 3rd cage in the second season is almost double that of the first season and similar differences are found for the grazed areas. For the rest of the cages, the differences are present but are much lower. The overall pattern suggests a moderate biomass growth in the first 5–6 months and much more rapid in the 2–3 that follow. This difference is depicted when all samples taken per period are summed and the average dry biomass production per day in the 3rd-period sampling is 12.1 kg/(day*ha) for the first season, compared to 5.8 and 5.2 for the 1st and 2nd periods respectively and 21.6 kg/(day*ha) for the 3rd period of the second season and 9.8 and 4.4 for the 1st and 2nd periods respectively (the differences are statistically significant for the second season, ANOVA $F_{(2,123)} = 10.850$, $p < 0.001$).

The overall biomass from all samplings (Figure 5) apart from differences between the two seasons for all sampling periods, depicts this pattern of biomass growth. The growth rate late in the season is much higher than that of the start and middle, even in the grazed areas. It is worth noting that the highest rates are recorded in both seasons in the 1st cage, which suggests that early grazing and then rest may provide higher overall productivity late in the season for phryganic pastures.

Grazing pressure reflects different practices of farmers that cannot always be attributed to the productivity of the particular pasture, but also to factors such as the proximity of the pasture to the shed (pastures located where the shed is are typically more heavily grazed) and the overall rotation of the flock during the grazing season in different pastures. Nevertheless, some of the more productive pastures are used very heavily (Table 4) and throughout the year, while for others the use is more seasonal.

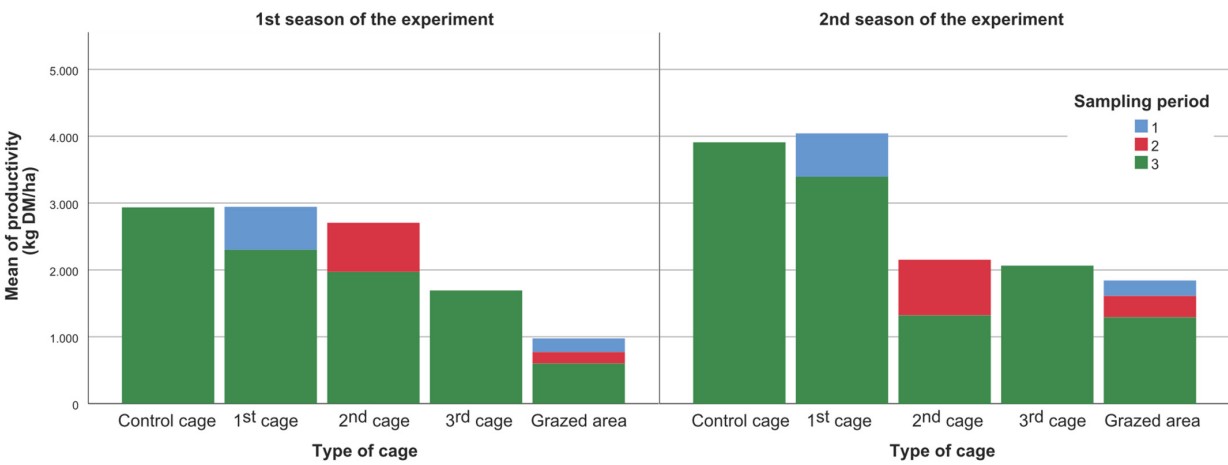

**Figure 5.** Sum of average biomass production per cage and sampling period for the two seasons.

**Table 4.** Grazing pressure ((animals/ha)/day) for each pasture.

| | Season | | | | Sampling Period | | | | | |
|---|---|---|---|---|---|---|---|---|---|---|
| | **1** | | **2** | | **1** | | **2** | | **3** | |
| **Pasture** | **Mean** | **Standard Error of Mean** | **Mean** | **Standard Error of Mean** | **Mean** | **Standard Error of Mean** | **Mean** | **Standard Error of Mean** | **Mean** | **Standard Error of Mean** |
| 1.1 | 6.35 | 0.97 | 4.14 | 0.68 | 8.58 | 0.77 | 6.65 | 0.95 | 3.36 | 0.44 |
| 1.2 | 8.41 | 2.57 | 13.69 | 3.12 | 2.34 | 1.35 | 2.59 | 0.24 | 17.92 | 1.55 |
| 1.3 | 1.47 | 0.41 | 2.05 | 0.26 | 3.20 | 0.05 | 2.21 | 0.21 | 1.01 | 0.19 |
| 1.4 | 13.39 | 0.78 | 12.93 | 0.86 | 9.63 | 0.44 | 12.09 | 0.06 | 15.00 | 0.28 |
| 2.1 | 6.57 | 0.58 | 4.03 | 0.47 | 6.78 | 0.15 | 3.70 | 0.06 | 5.36 | 0.75 |
| 2.2 | 48,00 | 3.43 | 38.66 | 1.38 | 43.62 | 5.69 | 33.02 | 1.06 | 47.34 | 2.08 |
| 2.3 | 46.72 | 1.18 | 25.69 | 5.65 | 43.3 | 2.96 | 49.72 | 0.92 | 27.96 | 5.43 |
| 3.1 | 7.90 | 2.49 | 0.83 | 0.55 | 1.88 | 1.08 | 0.78 | 0.45 | 6.80 | 2.45 |
| 3.2 | 7.09 | 3.29 | 12.31 | 2.27 | 0.43 | 0.25 | 21.09 | 1.22 | 8.86 | 2.06 |
| 3.3 | 10.96 | 1.98 | 9.15 | 1.54 | 1.17 | 0.04 | 11.14 | 0.46 | 13.18 | 0.71 |
| 4.1 | 56.17 | 5.95 | 69.47 | 3.22 | 53.71 | 2.74 | 85.11 | 0.00 | 57.55 | 3.95 |
| 4.2 | 9.14 | 2.59 | 13.04 | 3.81 | 0.00 | 0.00 | 2.82 | 0.44 | 18.84 | 1.59 |
| 4.3 | 1.80 | 0.37 | 0.98 | 0.20 | 1.96 | 0.17 | 0.00 | 0.00 | 1.72 | 0.25 |
| 4.4 | 1.05 | 0.37 | 1.03 | 0.33 | 0.00 | 0.00 | 0.00 | 0.00 | 1.88 | 0.14 |
| Season 1 total | 16.07 | 1.75 | | | 13.21 | 3.79 | 16.86 | 4,54 | 16.90 | 2.11 |
| Season 2 total | | | 14.86 | 1.74 | 12.01 | 3.34 | 16.13 | 4.54 | 15.49 | 2.20 |

The four farms differ in many aspects: the average daily herbage productivity varies significantly from 5.5 kg DM/ha of farm 1 (with mostly undisturbed phryganic pastures) to 25.5 kg DM/ha for farm 2 (with many improved flat pastures). Nevertheless, these differences level out when the kg of DM that are available to grazing animals per day of grazing are calculated to around 2 kg/animal/day (2.7 on farm one, 2.0 on farms two and three, and 1.5 on farm four).

### 3.3. Plant Species, Diversity, and Protein Content

The average number of species recognized is 6.8 per pasture, with a minimum of two·found in a grazed area sample in the second season and a maximum of 39 species found in a control cage in the first season (Tables A1 and A2). Overall, more species were found in the first season than in the second one (7.4 and 4.3 species respectively, while the differences are statistically significant ANOVA F(1,138) = 35.205, $p < 0.05$).

Differences between cages are not statistically significant, revealing similar levels of diversity even in grazed areas (average number of species/families for grazed areas at 14.5 in the first season and 8.3 in the second, slightly bigger than the overall average for all cages and higher than that of the first cage). Grasses accounted for 4.4 species on average and legumes for one species, while those of other families were 5.6 (Table 5).

**Table 5.** Comparison between grasses, legumes, and plants of other families per season, the counts represent the number of repetitions (samplings), 9 for each year (control cage 1, 1st and 2nd cage × 2 = 4, 3rd cage 1, grazed area 3), 18 in total for each pasture, 18 × 14 = 252 in total.

| Plant Category | Season | Count | Mean | Minimum | Maximum | Std. Deviation |
|---|---|---|---|---|---|---|
| | 1st | 126 | 5.60 | 0.00 | 18.00 | 3.11 |
| Grasses | 2nd | 126 | 3.23 | 0.00 | 11.00 | 2.27 |
| | 1st and 2nd | 252 | 4.41 | 0.00 | 18.00 | 2.96 |
| | 1st | 126 | 1.74 | 0.00 | 6.00 | 1.48 |
| Legumes | 2nd | 126 | 0.40 | 0.00 | 2.00 | 0.55 |
| | 1st and 2nd | 252 | 1.07 | 0.00 | 6.00 | 1.30 |
| | 1st | 126 | 6.81 | 2.00 | 24.00 | 4.62 |
| Other families | 2nd | 126 | 4.43 | 0.00 | 13.00 | 2.69 |
| | 1st and 2nd | 252 | 5.62 | 0.00 | 24.00 | 3.95 |

The analysis of the plant samples for nutritional characteristics such as crude protein, ash, and crude fiber shows a small increase in protein content as the season progresses: on average in the third cage 7.27% of the herbage dry matter was crude protein, compared to 5.6% in the control cage and 5.8% in the first, while in the grazed area the lowest content was measured at 5.4% (Table 6), but the differences are not statistically significant.

**Table 6.** Chemical analysis of herbage samples harvested during the first and second seasons, the counts of samples represent the samples taken. The analysis was made for the samples of the third sampling, 14 samples (one from each pasture) for each type of cage per session of the experiment, 14 × 2 = 28 in total. The counts of measurable samples represent the samples capable of being analyzed.

| Type of Cage | Count of Samples | Crude Protein | | | Crude Fiber | | | Ash | | |
|---|---|---|---|---|---|---|---|---|---|---|
| | | Count of Measurable Samples | Mean | Std. Deviation | Count of Measurable Samples | Mean | Std. Deviation | Count of Measurable Samples | Mean | Std. Deviation |
| Control cage | 28 | 26 | 5.60 | 1.77 | 27 | 32.72 | 3.18 | 27 | 12.02 | 2.91 |
| 1st cage | 28 | 25 | 5.77 | 1.31 | 25 | 32.96 | 2.60 | 26 | 11.34 | 1.81 |
| 2nd cage | 28 | 18 | 6.54 | 2.36 | 20 | 33.06 | 2.92 | 20 | 10.21 | 1.52 |
| 3rd cage | 28 | 14 | 7.27 | 2.18 | 14 | 31.29 | 1.99 | 14 | 10.70 | 2.70 |
| Grazed area | 28 | 23 | 5.49 | 2.13 | 22 | 33.14 | 2.68 | 23 | 10.25 | 1.68 |

## 4. Discussion

Western Lesvos is a typical semi-arid Mediterranean area, facing socio-economic and environmental issues [36], leading to soil degradation and desertification [47]. According

to estimations, large parts of the island are already degraded and under the process of desertification, including areas such as oak forests [40,48]. According to [49], the region is a typical case of the association between several socio-economic development policies and the environmental degradation process with practices such as overgrazing and deforestation, but the degree of dependency on sheep farming for livelihoods in many remote villages is high [38,49].

In our approach, we investigate the relationship between grazing practices, productivity, and plant diversity not at a specific time of the season (typically in the end in most of the literature, e.g., [50–52], but along the whole season [53]).

This allows a detailed view of the relationship between the characteristics of the pasture and the grazing practices of the farmers. Some of the most important findings are related to the pastures and their management, including grazing within the season, plant diversity, and long-term sustainability of the system and the pastures.

In terms of the production of pastures, the results of the significant seasonal changes in biomass production suggest that the late spring growth of grazed plants produces significantly more biomass when the temperatures of the soil and air rise owing to the rapid growth of many plants [54]. Therefore, pastures that can be left ungrazed for a short period of time during that period, have a better potential to yield biomass later in the season, although this cannot be extended over a long time due to the proximity of the dry summer. The overall productivity in the area is in the upper limits of the range suggested by the literature for semi-natural lands found on marginal soils (from $0.5$–$1.0$ t ha$^{-1}$ year$^{-1}$) [18], but the high heterogeneity of the productivity in our pastures suggests that those with production values lower than $0.5$ t ha$^{-1}$ year$^{-1}$ are on unfavorable soils while the others are in the range of agriculturally improved grasslands, though most of the latter are former fields of arable crops not more than 30–40 years ago. The ranges suggested by the literature for daily dry matter accumulation are within the values in our pastures as well [19,20].

In such an area, the long history of grazing and, in general, the use of the pastures has been suggested as important in understanding current trends [6]. The pastures we measured can be categorized into two categories: pastures that were arable fields until roughly the 1970s or 1980s when cultivation of arable plants more or less stopped on the western part of the island [38]. These pastures are in flat areas with deeper soils and their grazing histories are short compared to the rest of the area. The rest of the pastures are on sloping land, with shallower soils and long grazing history. Our findings show that despite important variability within the categories, there are significant differences in biomass production between these types of pastures that seem to reflect these histories, but also their differences in abiotic features, notably soil depth. Nevertheless, what also comes out is that these differences in productivity are not mirrored in the plant diversity data. The lack of a significant relationship between diversity and productivity has been also found in both experimentally [55–57] and naturally assembled Mediterranean herbaceous communities [58]. In general, productivity is considered to explain a small portion of the overall variation in grassland plant diversity worldwide [59,60].

Furthermore, the findings of the recognized species show that grazing seems to be unrelated to the overall level of diversity within each season. It has to be noted though that the small surfaces used in the approach may affect the overall number of species, especially in localities within each pasture where specific species may be found (e.g., in rocky places or along streamlines). The number of species is relatively low for these types of ecosystems [61,62], probably related to the long-term effects of heavy grazing on many of the pastures [63,64]. Another plausible explanation of the patterns of herbage growth can be offered on the basis that, after several decades of high-grazing pressure, flora in the area has been adapted to the specific biotic and abiotic pressures of the area. Indeed, the ability of plants to grow, reproduce, and survive under changing environmental conditions depends on their efficiency in acclimatizing and adapting, where the first one is associated with short-term challenges as opposed to the second one [65–67].

The nutritional composition of grazed material shows a low content of nutrients. The CP values are as low as cereal straw, but the crude fiber is somehow better than straw. However, a pattern of differences between the samplings within a year is associated with less mature herbage biomass harvested from the cages set later in the season (2nd and 3rd cage) which appear higher in CP and lower in crude fiber. This effect can also be associated with the higher presence of legumes in the 3rd cages, and their lower presence in samples from the control cages that represent plants grown very early in the season.

Overall, the approach followed here can be used for monitoring different aspects of the productivity of pastures in semi-arid and heavily grazed areas including seasonal differences and design improvement strategies.

## 5. Conclusions

Semi-arid areas of poor soils and intense relief have been grazed for millennia. Sustaining an important livelihood activity for local populations, while conserving pasture productivity and avoiding land degradation within climate change is a challenge that will determine socio-economic viability and environmental conservation in these areas. In this study, we analyzed pastures grazed by sheep in a semi-arid Mediterranean area, recording the land cover, herbaceous productivity, and plant diversity over the grazing season. Grazing history seems to be important and perhaps more important than seasonal grazing practices differences between pastures. This seems to suggest that the improvement and sustainability of pastures is a long-term concern and should be the result of a multi-annual plan that can provide guidance to farmers on how they have to treat their pastures. This is not easy, of course, within a climate of increasing livestock numbers and intensifying feeding practices, both in and off the pastures.

Past grazing practices that incorporated the cultivation of crops were possible and grazing after the end of the growing season and in the fallow years could provide an improvement plan that may prove more sustainable in the long term. Such policies are not incorporated into the current thinking and practice of the Common Agricultural Policy of the EU and a shift toward greater complementarity between cultivation and grazing and also towards less intensive grazing practices could provide incentives to farmers and local societies to continue a long-term and sustainable use of these pastures.

**Author Contributions:** Conceptualization, G.P., T.K., I.H. and P.G.D.; methodology, G.P., T.K., I.H. and P.G.D.; validation, G.P.; formal analysis, G.P., T.K. and P.G.D.; investigation, G.P.; resources, G.P. and T.K.; data curation, G.P. and T.K.; writing—original draft preparation, G.P., T.K. and I.H.; writing—review and editing, T.K., I.H. and P.G.D.; visualization, G.P.; project administration, T.K. All authors have read and agreed to the published version of the manuscript.

**Funding:** This research received no external funding.

**Data Availability Statement:** Data are available from the researchers on demand.

**Conflicts of Interest:** The authors declare no conflict of interest.

## Appendix A

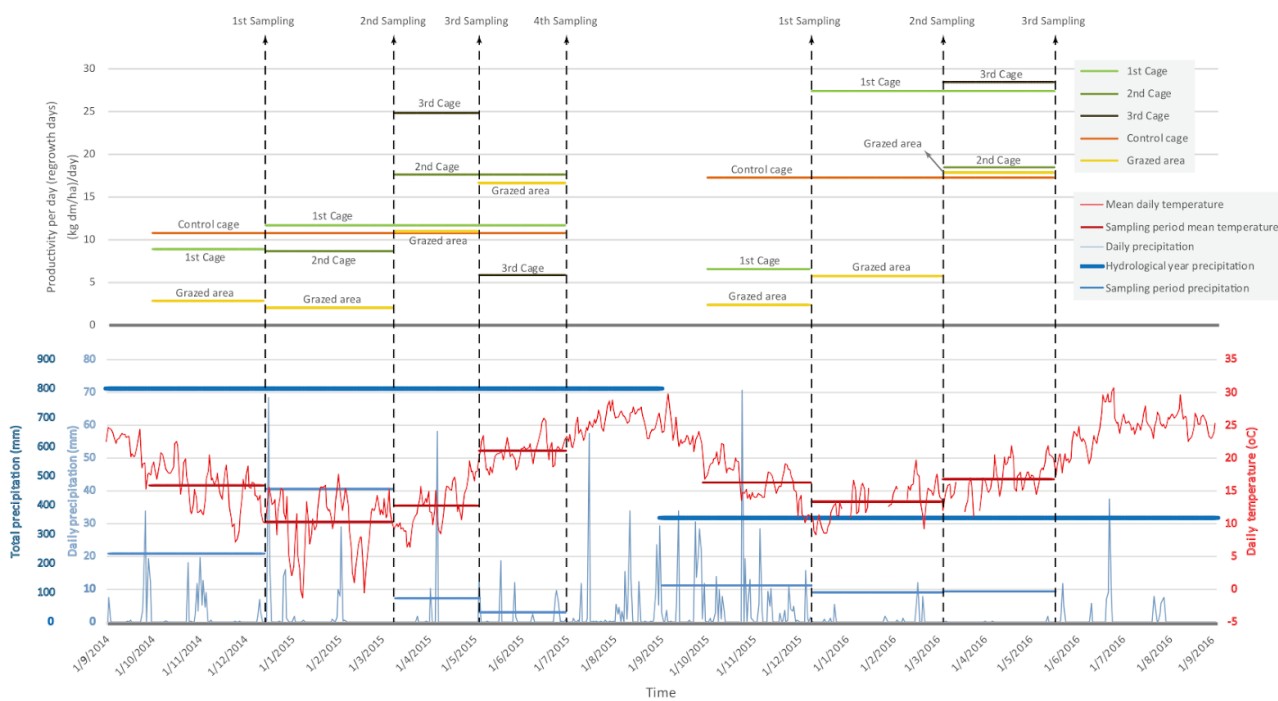

**Figure A1.** Productivity per day of regrowth for all cages and climate data during the experiment.

**Table A1.** The number of different recognized species per season and type of cage for all pastures.

| | Season 1 | | | | | Season 2 | | | | |
|---|---|---|---|---|---|---|---|---|---|---|
| Pasture | Control Cage | 1st Cage | 2nd Cage | 3rd Cage | Grazed Area | Control Cage | 1st Cage | 2nd Cage | 3rd Cage | Grazed Area |
| 1.1 | 16 | 15 | 15 | 11 | 25 | 11 | 8 | 9 | 11 | 14 |
| 2.1 | 18 | 22 | 23 | 23 | 30 | 6 | 15 | 15 | 11 | 13 |
| 2.2 | 6 | 13 | 6 | 9 | 9 | 5 | 8 | 4 | 4 | 5 |
| 1.2 | 5 | 12 | 8 | 8 | 7 | 4 | 3 | 9 | 11 | 4 |
| 3.1 | 14 | 12 | 16 | 18 | 12 | 7 | 5 | 4 | 5 | 7 |
| 4.1 | 14 | 14 | 6 | 8 | 9 | 5 | 9 | 6 | 8 | 2 |
| 4.2 | 9 | 9 | 11 | 21 | 8 | 3 | 3 | 4 | 5 | 5 |
| 1.3 | 23 | 7 | 15 | 15 | 16 | 23 | 10 | 11 | 7 | 8 |
| 3.2 | 13 | 8 | 11 | 16 | 10 | 5 | 4 | 8 | 6 | 6 |
| 2.3 | 10 | 9 | 11 | 13 | 11 | 5 | 5 | 5 | 6 | 8 |
| 4.3 | 39 | 22 | 25 | 18 | 29 | 15 | 14 | 15 | 17 | 21 |
| 1.4 | 10 | 8 | 10 | 5 | 5 | 13 | 8 | 4 | 8 | 6 |
| 3.3 | 17 | 28 | 20 | 37 | 18 | 10 | 7 | 8 | 13 | 12 |
| 4.4 | 12 | 13 | 6 | 4 | 15 | 7 | 7 | 4 | 5 | 5 |

**Table A2.** Plant families, the frequency they appeared during the samplings, and the number of different species recognized.

|   | Families | Number of Species |
|---|----------|-------------------|
| 1 | Poaceae | 19 |
| 2 | Compositae | 26 |
| 3 | Fabaceae | 17 |
| 4 | Plantaginaceae | 5 |
| 5 | Caryophyllaceae | 11 |
| 6 | Umbelliferae | 9 |
| 7 | Liliaceae | 4 |
| 8 | Boraginaceae | 3 |
| 9 | Xanthorrhoeaceae | 2 |
| 10 | Polygonaceae | 4 |
| 11 | Geraniaceae | 6 |
| 12 | Brassicaceae | 2 |
| 13 | Gentianaceae | 2 |
| 14 | Apiaceae | 2 |
| 15 | Ranunculaceae | 3 |
| 16 | Hyacinthaceae | 2 |
| 17 | Lamiaceae | 3 |
| 18 | Rosaceae | 2 |
| 19 | Amaranthaceae | 1 |
| 20 | Cruciferae | 1 |
| 21 | Orobanchaceae | 1 |
| 22 | Malvaceae | 1 |
| 23 | Scrophulariaceae | 2 |
| 24 | Caprifoliaceae | 1 |
| 25 | Lythraceae | 1 |
| 26 | Oxalidaceae | 1 |
| 27 | Primulaceae | 1 |
|  | Total | 132 |

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
