# Peer review of "Grazing Land Productivity, Floral Diversity, and Management in a Semi-Arid Mediterranean Landscape"

_sustainability, doi:10.3390/su14084623_

Round 1

Reviewer 1 Report

General:

This is an interesting study. The work appears to be well-executed and the manuscript demonstrates a good command of relevant literature. The context is well-explained.

There are, however, some improvements that could be made. The most important improvements must focus on providing a clearer description of the experimental design. Without a clear description, it is difficult to assess the interpretation of the results and accuracy of the conclusions.

Specific points are:

1) The aims on page 3 need to be stated more explicitly. What is measured? What is being shown here?

2) The experimental design on pages 5 and 6 needs to be explained more clearly. Figure 2 is not sufficiently clear or adequately explained in the text. I found it confusing. 

For example, the first two cages comprise the Control and a second cage which is called the "1st cage". This "1st cage" was harvested three months later and was replaced by another cage. This new cage is also called the "1st cage". 

3) It is difficult to see how Figure 2 (horizontal time axis) relates to Figure 3 (vertical time axis).  I recommend that they have 'time' on the same axis. In addition, the number of columns (cages) in Figure 2 is different from the number of rows (cages) in Figure 2. 

4) In Table 2, the count (n) = 18, but in the site description it says there are 16 fields. Is this an error?

5) I do not understand where the counts (sample sizes, n) in Tables 3, 5 and 6 come from.  How does this relate to the experimental design?

6) Titles of all Tables and Figures need to include more detail (e.g. units stated consistently in all Tables and Figures, and labels giving 'days' in Figure 2 etc.). 

7) To help interpret the productivity data the reader needs to know which pastures were 'improved' and 'undisturbed'. This is not clear in the Tables and Figures, but the effects of improvement are described in the text of the Results. It is not possible to interpret the evidence in the Tables and Figures. 

8) Overall, interpretation of the seasonal variation, and response to rainfall, is hard to visualise

Author Response

Response to Reviewer 1 Comments

General:

Comments of Reviewer 1

This is an interesting study. The work appears to be well-executed and the manuscript demonstrates a good command of relevant literature. The context is well-explained.

There are, however, some improvements that could be made. The most important improvements must focus on providing a clearer description of the experimental design. Without a clear description, it is difficult to assess the interpretation of the results and accuracy of the conclusions.

Response

Thank you very much for these comments! We have done our best to meet all your comments in the revised manuscript. The experimental design has been described in more detail in the revised manuscript.

Specific points are:

Point 1: The aims on page 3 need to be stated more explicitly. What is measured? What is being shown here?

Response 1:

Thank you for the comment. In the revised text, we have stated more clearly the objectives of the paper, clarifying what we measure and what is presented in the paper.

Point 2: The experimental design on pages 5 and 6 needs to be explained more clearly. Figure 2 is not sufficiently clear or adequately explained in the text. I found it confusing. For example, the first two cages comprise the Control and a second cage which is called the "1st cage". This "1st cage" was harvested three months later and was replaced by another cage. This new cage is also called the "1st cage".

Response 2:

Thank you for the comment. In the revised text, we have clarified the experimental design and provided more information on the different cages. We explain that indeed “cage 1” is harvested after 3 months and then put back in its place and harvested again at the end of the season. So, we have two harvests of “cage 1” and of “cage 2”. This design ensures a glimpse into regrowth at different time periods. The revised text now reads:

“The cages were made with wire mesh (eye 1x1.2 cm), covering a circular area of 0.25 m2 and with a height of 0.7 m covered with a wire mesh lid. To anchor the cages, three iron rods were nailed to the ground. The first two cages were placed at the start of the vegetative period, in October (Figures 2 and 3). Οne of them (“control cage”) was sampled only at the end of the growing period (control cage). The second (“1st cage”) was harvested at ground level three months later (December) and a second time at the end of the season along with all cages, while a new cage (“2nd cage”) was placed nearby. During the second sampling, in early March, the 2nd cage, placed in December, was harvested and a new cage (“3rd cage”) was also placed. During the third sampling, in late May - early June, when herbage was almost dry, all four cages were harvested (the “1st cage” and the “2nd cage” for the second time). To record the standing herbaceous biomass in the plot during each sampling session, a sample was harvested from an area of 0.25m2 representative of the overall field vegetation (“grazed area sample”) (43). The final sampling was made at different periods each year due to the weather conditions that matured plants earlier in the second year: in the first year, this was in early June, while in the second in mid-May.”

Point 3: It is difficult to see how Figure 2 (horizontal time axis) relates to Figure 3 (vertical time axis).  I recommend that they have 'time' on the same axis. In addition, the number of columns (cages) in Figure 2 is different from the number of rows (cages) in Figure 2.

Response 3:

The figures have been harmonized. The different number “3rd cage” is mentioned, refers to the fact that in the second year of samplings, the final sampling had to be done earlier that the first year due to the weather conditions (plants matured approximately 20 days earlier in the second year). We standardized these samplings and this is now mentioned in the revised text.

Point 4: In Table 2, the count (n) = 18, but in the site description it says there are 16 fields. Is this an error?

Response 4:

No, this is not an error, this figure represents the number of repetitions (samplings), which are 9 for each year (control cage X 1 =1, 1st 2nd cage X 2 = 4, 3rd cage X1=1, grazed area X 3=3, in total 9), therefore 18 in total. This is now explained in the revised caption of Table 2 and Figure 2.

Point 5:  I do not understand where the counts (sample sizes, n) in Tables 3, 5 and 6 come from.  How does this relate to the experimental design?

Response 5:

The counts in Tables 3 and 5 represent the number of repetitions (samplings), 9 for each year (control cage 1, 1st and 2nd cage X 2 = 4, 3rd cage 1, grazed area 3), 18 in total for each pasture, 18 X 14 pastures = 252 in total. This is now explained in the revised captions of Tables 3, 5 and Figure 2. In the revised Table 6 the counts of samples represent the samples taken. The chemical analysis was made for the samples of the third sampling, 14 samples (one from each pasture) for each type of cage per session of the experiment, 14 X 2 = 28 in total. The counts of measurable samples represent the samples capable of being analyzed.

Point 6:   Titles of all Tables and Figures need to include more detail (e.g. units stated consistently in all Tables and Figures, and labels giving 'days' in Figure 2 etc.).

Response 6:

All Table titles have been revised.

Point 7: To help interpret the productivity data the reader needs to know which pastures were 'improved' and 'undisturbed'. This is not clear in the Tables and Figures, but the effects of improvement are described in the text of the Results. It is not possible to interpret the evidence in the Tables and Figures.

Response 7:

The numbers and codes of the improved pastures are mentioned now in the revised text (research methods).

Point 8: Overall, interpretation of the seasonal variation, and response to rainfall, is hard to visualise

Response 8:

Response to rainfall is hard to visualize because we had rainfall information for the whole area and not for individual pastures, while the variability of seasonal variations in regrowth was so significant that it was not feasible to meaningfully correlate rainfall with plant growth. Instead, we have used a more qualitative approach in the analysis, visualization and discussion.

Reviewer 2 Report

This paper research the use, productivity and flora diversity of typical Mediterranean grazing using field data. I think the materials used in this manuscript are solid and valuable. And the results are somewhat interesting. However, the contribution of the paper is not very clear, that is, what is the knowledge revealed by the data, rather than a simple description. The introduction too redundant, so it is suggested to simplify it.

Author Response

This paper research the use, productivity and flora diversity of typical Mediterranean grazing using field data. I think the materials used in this manuscript are solid and valuable. And the results are somewhat interesting. However, the contribution of the paper is not very clear, that is, what is the knowledge revealed by the data, rather than a simple description. The introduction too redundant, so it is suggested to simplify it.

The article adopts a sound methodology in order to study the relationship  between grazing practices and soil features and productivity particularly with the aim to cope with desertification processes fuelled , in semi-arid  sud mediterranean regions by climate change trends.

In doing so the article adopts a sound  methodology that allow to appraise,  and focus on,  key insight about the necessity to consider a  multi annual perspective in grazing practices. That especially valuing long-term  and practically acquired knowledge  about the relevance of an integrated form of farming/grazing patterns, relating both to seasonality and geo-pedological conditions.

Thank you very much for these comments! We hope that the revised version of the paper is even better.

Reviewer 3 Report

The article adopts a sound methodology in order to study the relationship  between grazing practices and soil features and productivity particularly with the aim to cope with desertification processes fuelled , in semi-arid  sud mediterranean regions by climate change trends.

In doing so the article adopts a sound  methodology that allow to appraise,  and focus on,  key insight about the necessity to consider a  multi annual perspective in grazing practices. That especially valuing long-term  and practically acquired knowledge  about the relevance of an integrated form of farming/grazing patterns, relating both to seasonality and geo-pedological conditions.

In such a prospect the article provides remarkable and steady findings to suggest different approaches in rural development  actions supported and funded by EU policies and to appraise the pivotal importance of low scale and multifucntional farming schemes. The latter  as suggesting a long-term  featured co-evolutionary approach between anthropoghenic actions and ecosystems, and to correct intesive and monofunctional activities that turns out as not being enough resilient to climate change and  desertification impacts.

Author Response

In such a prospect the article provides remarkable and steady findings to suggest different approaches in rural development  actions supported and funded by EU policies and to appraise the pivotal importance of low scale and multifucntional farming schemes. The latter  as suggesting a long-term  featured co-evolutionary approach between anthropoghenic actions and ecosystems, and to correct intesive and monofunctional activities that turns out as not being enough resilient to climate change and  desertification impacts.

Thank you very much for these comments! We hope that the revised version of the paper is even better.

We remain at your disposal for further comments and clarifications.

Sincerely, the authors

Round 2

Reviewer 1 Report

The corrections address the points I raised in my review. 

I have attached a file which contains minor edits to improve the English.

Author Response

Thank you very much for these comments! We remain at your disposal for further comments and clarifications.

Reviewer 2 Report

I think the revision has revised well.

Author Response

Thank you very much, we remain at your disposal.    Sincerely, the authors.